# Analysis of Cross-Combinations of Feature Selection and Machine-Learning Classification Methods Based on [^18^F]F-FDG PET/CT Radiomic Features for Metabolic Response Prediction of Metastatic Breast Cancer Lesions

**DOI:** 10.3390/cancers14122922

**Published:** 2022-06-14

**Authors:** Ober Van Gómez, Joaquin L. Herraiz, José Manuel Udías, Alexander Haug, Laszlo Papp, Dania Cioni, Emanuele Neri

**Affiliations:** 1Nuclear Physics Group and IPARCOS, Faculty of Physical Sciences, University Complutense of Madrid, CEI Moncloa, 28040 Madrid, Spain; obervang@ucm.es (O.V.G.); jlopezhe@ucm.es (J.L.H.); jose@nuc2.fis.ucm.es (J.M.U.); 2Academic Radiology and Master in Oncologic Imaging, Department of Translational Research, University of Pisa, Via Roma, 67, 56126 Pisa, Italy; dania.cioni@med.unipi.it; 3Division of Nuclear Medicine, Department of Biomedical Imaging and Image Guided Therapy, Medical University of Vienna, 1090 Vienna, Austria; alexander.haug@meduniwien.ac.at; 4Center for Medical Physics and Biomedical Engineering, Medical University of Vienna, 1090 Vienna, Austria; laszlo.papp@meduniwien.ac.at; 5Italian Society of Medical and Interventional Radiology, SIRM Foundation, Via della Signora 2, 56122 Milan, Italy

**Keywords:** FDG-PET/CT, metastatic breast cancer, feature selection, machine learning, radiomics, metabolic response, heterogeneous metabolic responses

## Abstract

**Simple Summary:**

Breast cancer is a leading cause of morbidity and mortality worldwide. The metastatic disease is largely responsible for cancer patient deaths, and its treatment implies usually different therapies. In this context, the prediction of response to treatment or depiction of treatment-resistant phenotypes is essential in clinical practice, especially in the new era of precision medicine. In this study, we used several combinations of feature selection methods and machine-learning classifiers to construct predictive models of the metabolic response to the treatment of metastatic breast cancer lesions. These models were based on clinical variables and radiomic features extracted from 2-deoxy-2-[^18^F]fluoro-D-glucose positron emission tomography/computed tomography ([^18^F]F-FDG PET/CT) images, obtained prior to the treatment. Our main goal was to know if this prediction was feasible and to identify those combinations with better predictive performance. We found that several combinations were successful to predict the metabolic response to treatment, of which the least absolute shrinkage and selection operator (Lasso) + support vector machines (SVM) had the best mean performance in terms of area under the curve, in both training and validation cohorts. Model performances depended largely on the selected combinations.

**Abstract:**

Background: This study aimed to identify optimal combinations between feature selection methods and machine-learning classifiers for predicting the metabolic response of individual metastatic breast cancer lesions, based on clinical variables and radiomic features extracted from pretreatment [^18^F]F-FDG PET/CT images. Methods: A total of 48 patients with confirmed metastatic breast cancer, who received different treatments, were included. All patients had an [^18^F]F-FDG PET/CT scan before and after the treatment. From 228 metastatic lesions identified, 127 were categorized as responders (complete or partial metabolic response) and 101 as non-responders (stable or progressive metabolic response), by using the percentage changes in SULpeak (peak standardized uptake values normalized for body lean body mass). The lesion pool was divided into training (n = 182) and testing cohorts (n = 46); for each lesion, 101 image features from both PET and CT were extracted (202 features per lesion). These features, along with clinical and pathological information, allowed the prediction model’s construction by using seven popular feature selection methods in cross-combination with another seven machine-learning (ML) classifiers. The performance of the different models was investigated with the receiver-operating characteristic curve (ROC) analysis, using the area under the curve (AUC) and accuracy (ACC) metrics. Results: The combinations, least absolute shrinkage and selection operator (Lasso) + support vector machines (SVM), or random forest (RF) had the highest AUC in the cross-validation, with 0.93 ± 0.06 and 0.92 ± 0.03, respectively, whereas Lasso + neural network (NN) or SVM, and mutual information (MI) + RF, had the higher AUC and ACC in the validation cohort, with 0.90/0.72, 0.86/0.76, and 87/85, respectively. On average, the models with Lasso and models with SVM had the best mean performance for both AUC and ACC in both training and validation cohorts. Conclusions: Image features obtained from a pretreatment [^18^F]F-FDG PET/CT along with clinical vaiables could predict the metabolic response of metastatic breast cancer lesions, by their incorporation into predictive models, whose performance depends on the selected combination between feature selection and ML classifier methods.

## 1. Introduction

Breast cancer is the most diagnosed cancer and the second leading cause of death for cancer among women worldwide, surpassed only by lung cancer [1]. After the diagnosis, it is important to define accurately the initial extent of the disease because it will affect the treatment election. For instance, patients with locally advanced stages could receive neoadjuvant chemotherapy, followed by a breast operation and adjuvant radiotherapy, and depending on the hormonal receptor status of the tumor, they could receive posteriorly an adjuvant targeted or hormonal therapy [2]. However, after the primary treatment, some percentage of patients will have a recidive months, or even years, after the initial diagnosis and treatment. This recidive could be locoregional or distant (metastatic disease). The metastatic disease is largely responsible for most cancer patient deaths, and its treatment implies usually the use of different systemic treatments including chemotherapy, hormonal therapy, targeted therapies, and immunotherapy, with or without local treatments such as radiation therapy. Usually, these therapies have several side effects, which could be avoided by considering changing or discontinuing the therapeutic regimen. In this context, the prediction of response to treatment or depiction of treatment-resistant phenotypes is essential in clinical practice, especially in the new era of precision medicine [3].

Currently, [^18^F]F-FDG PET/CT is widely used in oncology and is usually performed in breast cancer patients for recurrence detection or treatment response assessment, both in the context of neoadjuvant therapy or metastatic cancer treatment [4]. Usually, only one lesion is assessed after the neoadjuvant treatment, whereas in patients with metastatic breast cancer, several target lesions can be selected (others may not be considered) for assessing after the treatment, obtaining finally a global clinical response. Changes in tumor metabolic activity, assessed by [^18^F]F-FDG PET/CT, are an early indicator of treatment effectiveness for breast cancer, in both neoadjuvant and metastatic settings [5]. Likewise, a significant reduction in the metabolic activity of the tumor (i.e., treatment-sensible tumors), early during therapy, is associated with longer overall survival and progression-free survival in this tumor [6]. However, in metastatic disease, the post-therapeutic response assessment frequently shows heterogeneous responses, which means that responding and nonresponding metastatic lesions coexist (dissociated response) [7]. Huyge et al. [8] showed intraindividual variability in the [^18^F]F-FDG PET/CT metabolic response among lesions of bone-dominant metastatic breast cancer patients treated with different systemic therapies (chemotherapy, hormone therapy, and anti-HER2 targeted therapy). This dissociated metabolic response tended to be associated with a better outcome than homogeneous nonresponse, which may suggest that in the case of dissociated disease, the prognosis will depend on the number, the localization, and the intrinsic aggressivity of the progressing lesions [7]. Shortly after initiating systemic treatment, [^18^F]F-FDG PET/CT provides a whole-body quantitative assessment of treatment-induced changes in tumoral glycolysis, allowing an early evaluation of the response to the treatment on a lesion-by-lesion basis. Thus, lesions with discordant evolution within the whole-tumor burden could be rapidly detected and reveal a heterogeneous response.

On the other hand, even when simple metabolic parameters derived from the ^18^F-FDG-PET/CT study have shown to be valuable to predict the treatment response in breast cancer and other malignant tumors, in recent years, there is an increasing interest in the clinical and prognostic utility of quantitative-imaging analysis through radiomics [3,9]. Radiomics refers to the extraction and analysis of quantitative-imaging features extracted from medical images, such as CT, PET, magnetic resonance (MR), and others, which have been shown to reflect mechanisms occurring at genetic and molecular levels [10]. From this point of view, radiomics could find patterns in medical images that allow us to detect disease, understand the pathological process, or predict the medical evolution of patients. Specifically, in oncology, radiomic features relating to tumor size, shape, voxel intensities, and texture allow tumor characterization (radiomic signature) and have shown their ability for diagnosing and predicting in several malignancies [11,12]. In breast cancer, most radiomic studies have been carried out with MR images in a neoadjuvant treatment context [13]. However, some studies appearing more recently have explored the potential of radiomics with PET/CT, but none was performed in patients with metastatic breast cancer [14], which makes this study much more relevant. Recently, image-based biomarkers designed by using radiomic features are becoming more frequently implemented with machine-learning (ML) classifiers, which have shown great promise [15]. This approach relies on a pipeline, including extraction of numerous handcrafted imaging features, followed by feature selection and machine-learning-based classification. Feature selection or reduction variable methodologies are performed before any ML predictive model construction, because redundant and irrelevant features are thus removed from further analysis, improving the ML classifier performance [16]. We hypothesize that predictive models consisting of combinations of feature selection methods and ML classifiers, being fed by clinical and radiomic features extracted from [^18^F]F-FDG PET/CT, can adequately predict the metabolic response to the treatment of individual metastatic breast cancer lesions, allowing the creation of quantitive indices for the heterogeneous response to the treatment.

## 2. Materials and Methods

### 2.1. Patient Cohort

We retrospectively revised the clinical chart of 136 patients with pathologically confirmed metastatic breast cancer at Vienna General Hospital between 2010 and 2015, and those who received chemotherapy, hormonal therapy, targeted therapies, or immunotherapy, with or without local radiotherapy, and underwent a baseline and follow-up [^18^F]F-FDG PET/CT to assess treatment response, were included in this study. Patients with other known disseminated malignancies were excluded. Clinical variables at initial diagnosis, such as age, side of breast affectation, and tumor TNM stage according to the seventh edition of the *AJCC Cancer Staging Manual* [17] were recorded. Additionally, we registered information about the pathologic biomarkers (i.e., estrogen and progesterone hormone receptor status, human epidermal growth factor receptor 2 (HER2) and, P53 and Ki-67 protein expression intensity, as cellular markers for proliferation) for both primary and metastatic lesions. Each pathologic biomarker was labeled with an integer between 0 and 3, according to the pathologic intensity informed. Thus, the differences in the pathological biomarker’s intensity between primary and metastatic lesions were scored. For instance, a patient with progesterone receptor positivity of 3 in the primary tumor, and documented progesterone receptor negativity in at least one of the metastatic lesions, woud be recorded as a progesterone receptor variation (∆) of −3. In addition, the treatment received for each patient, as well as the time between the baseline and follow-up [^18^F]F-FDG PET/CT, was registered.

### 2.2. PET/CT Image Acquisition

All patients fasted at least 5–6 h before an intravenous administration of a dose of approximately 200–350 MBq of 18F-FDG (body-weight-adapted). After an uptake time of around 60 min, a whole-body [^18^F]F-FDG PET/CT from mid-cranium to the upper thigh was performed using a 64-row multi-detector PET/CT system (Biograph 64 TruePoint; Siemens, Erlangen, Germany) with an axial field-of-view of 216 mm, a PET sensitivity of 7.6 cps/kBq, and a transaxial PET resolution of 4–5 mm (full-width at half-maximum, FWHM). Finally, most of the images had a voxel size of 4.07 × 4.07 × 3.00 mm^3^.

### 2.3. ROI Delineation

For each patient, one or several metastatic lesions were included. One experienced nuclear medicine physician reviewed all baseline [^18^F]F-FDG PET/CT images, considering target lesions or any pathological [^18^F]F-FDG –avid lesion with or without a corresponding anatomic lesion on the CT scan and suggestive of metastasis (lesions could be in any organ, lymph-node-inclusive). Posteriorly, these lesions were assessed on the follow-up [^18^F]F-FDG PET/CT images. For both basal and follow-up studies, the delineation process was performed using the Hermes Hybrid 3D software, version 2.0 (Hermes Medical Solutions, Stockholm, Sweden). First, a cuboid volume of interest (VOI, 5 × 5 × 5 voxels) was defined in the mediastinum to serve as a background reference. Then, the target lesions, visualized in the PET images, were delineated using a semiautomatic region-growing tool to generate corresponding lesion VOIs-PET, where only voxels with values higher than the mean of the background were included. These VOIs were dilated by 5 voxels by an automated dilatation tool. This step was performed to avoid interpolation artifacts at border voxel positions in the VOIs during the resampling. Finally, they were used as an image mask to obtain VOIs-CT [18].

### 2.4. Metabolic Parameters Extraction

From the PET VOIs, metabolic parameters such as maximum, mean, minimum, and peak standardized uptake values (SUVmax, SUVmean, SUVmin, and SUVpeak, respectively), as well as metabolic tumor volume (MTV), total lesion glycolysis (TLG), and SULpeak were obtained, where the SULpeak is defined as the average SUV within a 1 cm^3^ spherical VOI, centering around the hottest point in the tumor, and corrected by the lean body mass of the patient.

### 2.5. Image Preprocessing

VOIs-PET voxel intensities were converted to standardized uptake values (SUVs) based on body weight; thus, several common metabolic parameters were extracted. VOIs-CT voxel intensities had Hounsfield units. Before performing any image feature extraction, the VOI intensities were discretized by employing a fixed bin number of 64 [19]. It reduces the effect of noise in radiomics analysis by changing the substantially continuous voxel intensity to a discontinuous value and speeding up texture feature calculation. To remove possible individual acquisition differences, and thus maintain the reproducibility, PET and CT images were resampled into a voxel size of 1 × 1 × 1 mm^3^ by cubic interpolation [20,21]. Further image preprocessing was not performed.

### 2.6. PET/CT Response Assessment

The baseline and follow-up SULpeak values of each target lesion allowed obtaining the percentage changes in SULpeak as:ΔSULpeak = (SULpeak[follow-up] − SULpeak[baseline]) SULpeak[baseline]

If the target lesion was not present in the follow-up [^18^F]F-FDG PET/CT study, the follow-up SULpeak was considered as the background. Following the PET response criteria in solid tumors (PERCIST) [22], the individual metabolic response of each metastatic lesion was categorized as:Complete metabolic response (CMR): the disappearance of the metabolically active lesion;Partial metabolic response (PMR): more than 30% decrease in SULpeak;Progressive metabolic disease (PMD): more than 30% increase in SULpeak;Stable metabolic disease (SMD): does not meet the above criteria.

Lesions with CMR and PMR were considered responders, while lesions with SMD and PMD as nonresponders. The assessment of the global response, as proposed by PERCIST, was not performed. For this, the greatest ^18^FDG-avid lesions are measured (up to 5 lesions, maximum 2 lesions per organ) and summed to determine the metabolic change. Thus, the global response to therapy can be assessed as a percent change in the SULpeak of a single lesion (not necessarily the same lesion) or the sum of the lesions.

### 2.7. Radiomic Features Extraction

From the baseline VOIs-PET and VOIs-CT, radiomic features were extracted in the platform MATLAB by adapting an open-source radiomic analysis package [23], which follows the definitions of features from the imaging biomarker standardization initiative (IBSI) [21]. An overview of the radiomic process implemented in this article can be seen in Figure 1. The radiomic features’ name and their mathematical expressions are summarized in Appendix A.

#### Texture Features

A total of 101 textural features were extracted for each contoured tumor on both [^18^F]F-FDG PET and CT images, respectively, i.e., 202 textural features in total. Of these 101 features, 13 were intensity histogram features and 88 were textural features (of which 31 explore intratumoral heterogeneity).

### 2.8. Univariate Statistical Analysis

Univariate analysis was performed to investigate associations of single features with the outcome (metabolic response or not). Firstly, all features were normalized via z-score normalization to zero mean and unit variance. Then, for each clinical variable or image feature, a chi-squared (noncontinuous variables) or Mann–Whitney U statistic test (for continuous variables) was calculated. This statistical analysis was performed by using IBM SPSS Statistics for Windows, Version 25.0. Armonk, NY, USA: IBM Corp. The DeLong test was used to statistically compare the AUCs between the models. The significance level was chosen as a two-tailed *p* < 0.05.

### 2.9. Machine-Learning Model

#### 2.9.1. Feature Selection

Radiomics studies usually have hundreds of features, many of which are highly correlated; this makes necessary feature selection methods to avoid collinearity, reduce dimensionality, minimize noise, and avoid overfitting problems [16]. Hence, initially, a data-preprocessing methodology was implemented to reduce the large set of features to a subset of the most significant features. A pairwise Spearman correlation matrix was used to identify pairs of highly correlated features (|r| ≥ 0.90). Finally, from each pair, only those with the best association to the target variable (responders or not) were retained. After this data-preprocessing step, seven popular feature selection methods were used to further reduce the number of features: analysis of variance (ANOVA)-F-test (AFT), mutual information (MI), least absolute shrinkage and selection operator (Lasso), Wilcoxon test (WT), hierarchical clustering (HC), principal component analysis (PCA), and independent component analysis (IPA). These methods were chosen because of their popularity in several publications on radiomics and machine-learning methods [24,25,26].

#### 2.9.2. Classification Methods

To classify tumor lesions into responders and nonresponders (0 or 1), we investigated seven popular machine-learning classifiers: support vector machines (SVM), random Forest (RF), Gaussian naive Bayes (GNB), logistic regression (LR), k-nearest neighborhood (KNN), adaptative boosting (AdaBoost), and neural network (NN) [27,28]. The acronyms for each feature selection method and ML classifier are listed in Table 1.

The feature selection and classification methods were implemented by using SciKit Learn package in python (scikit-learn version 0.21, python version 3.6.3) and using the open platform Google Colaboratory. Each of the seven feature selection methods was combined with all the seven classifiers, and each classifier was combined with all the seven feature selection methods, yielding 49 cross-combinations.

#### 2.9.3. Model Construction

To create the predictive radiomic-based models, we followed the next steps, which are recommended to perform a suitable model performance [29]:Data imputation by filling the empty data with a strategy of most frequent.Data splitting with a ratio (80:20) into X_*train*, X_*test*, y_*train*, and y_*test*, where X and y are the predictive features (clinical and radiomic features) and the target variable (responders or nonresponders), respectively. Only the training set was used to construct the models, and the test set for validation purposes.A synthetic minority oversampling technique (SMOTE) [30] was performed for oversampling the nonresponder to have the same number of instances as the responder in the training procedure.Data standardization: all variables are obligated to have a mean zero and standard deviation of one.The feature selection method, hierarchical clustering, was applied directly after the data-preprocessing methodology, to obtain a smaller number of features. However, ANOVA F-test, MI, PCA, IPA, Lasso, and Wilcoxon were initially coupled to each one of the seven ML classifier methods, and subsequently, an iterative process was implemented to find a subgroup of features with the best performance in terms of ACC and AUC. For them, curves of the number of features selected versus model performance were obtained, allowing for optimization of the final number of chosen features, i.e., to find the smaller number of features with only a small change in the model performance concerning the maximal (only changes < 0.05 were allowed if there were a significant reduction in the number of features). Additionally, we obtained a ranking of the features (i.e., the feature importance) for each cross-combination. For the feature selection method Lasso, a cross-validated estimation of the best alpha parameters was performed, using the mean squared error as cross-validation score, where higher values are better than lower values (Appendix A).ML classifier hyperparameter tuning was also performed through cross-validation, and by using the class GridSearchCV of SciKit Learn. For GNB, the default hyperparameter setting was used.Finally, the 49 cross-combinations (each one with a specific subset of features, and an ML classifier with specific hyperparameters) were trained using the training cohort.

#### 2.9.4. Model Performance Metrics and validation

The performance of the models was assessed by ROC analysis using the ACC and AUC metrics. Initially, 10-fold cross-validation was carried out in the training group; it splits the data into 10 equal parts and uses 9 parts for training and the rest for testing. The feature selection methods were already included in the cross-validation algorithm so that their contribution to the final model was reflected in the performance metrics. For each fold, ROC curves were generated; in addition, for the whole 10-fold, the mean AUC and ACC were computed. This cross-validation allows some hyperparameter tuning before the validation of the models. After the model training, a validation of this was performed on the test cohort, also obtaining ROC curves and AUC and ACC for each model. We compared the ROC curves of the four models with the highest AUC. For this, the “pROC” package (using DeLong’s test) was used [31].

## 3. Results

### 3.1. Clinical Characteristics

Forty-eight patients were identified to have a biopsy-proven recidive and available pretreatment and follow-up [^18^F]F-FDG PET/CT. A total of 228 tumor lesions were visualized on the pretreatment PET/CT and followed up on the subsequent one. Patient and tumor characteristics are summarized in Table 2. The mean time elapsed between the initial and response PET/CT was 149 ± 70 days. A description of the treatment of each patient and places of affectation is given in Table 3.

In total, 72 lesions showed a complete metabolic response, 55 partial responses, 7 stable diseases, and 94 progressions. By considering as responders those lesions with CMR or PMR, and as nonresponder lesions with SMD or PMD, 127 were considered as responders and 101 as nonresponders, respectively.

### 3.2. Feature Extraction and Correlation

A total of 202 radiomic features (101 for each imaging modality) and 20 clinical and metabolic ones were obtained and investigated in terms of their association with the metabolic response of the patients. From these 222 features, the data-preprocessing filter removed 116 highly correlated ones, leaving a set of only 106 predictors. Figure 2 shows the heatmap of the feature and clinical variable correlations before and after the preprocessing filter application. It is interesting to note that clinical variables have a low Spearman correlation between them. By filtering, metabolic variables as MTV and TLG were removed because of their high correlation with SUVmax. The results of the univariate analysis are presented in Appendix A. Clinical variables such as tumor size, estrogen, progesterone receptor positivity, HER2, and tumor grade, as well as some metabolic variables such as SUVpeak, SUVmean, and SUVmax, had a statistically significant association with the target variable (metabolic response). Similar behavior was observed for various PET and CT image characteristics.

### 3.3. Feature Reduction

After applying feature reduction with HC, the initial 222 features were reduced to only 28. For AFT, MI, PCA, ICA, Wilcoxon, and Lasso classifiers, curves of performance versus the number of selected features were obtained for each combination between these selection methods and the ML classifiers, taking the number of features with better performance. More specifically, for each combination, we iteratively increased the number of selected features or components (for PCA and ICA) that finally fed the ML classifier, which was subsequently trained, and its performance assessed in each step through cross-validation. Figure 3 shows an example of how the classification performance (AUC) for Lasso + SVM and Lasso + RF changes according to the number of selected features. Maximal AUC and ACC are obtained with 14 features. Alternatively, we explored obtaining an optimized number of features, i.e., a smaller number of features but with only a small change concerning the maximal value already found. It was performed with 13 features. Table 4 shows the selected features for predicting the metabolic response of models Lasso + SVM, Lasso + NN, and MI + RF. They are ranked from left to right, where predictors on the left are of greater predictive significance. Appendix A shows a full list of selected features for Lasso in combination with the seven ML classifiers. Lasso allows sorting directly the features according their predictive importance, and the classifier only determines how many of them (with the highest importance) achieve the best performance. Thus, because both, Lasso + SVM and Lasso + NN had the best performance with 14 features, both models also had the same feature ranking ordering.

### 3.4. Cross-Validation

We examined 49 different combinations of feature selection and classification methods. Table 5 reports the performance values, in terms of mean AUC and its standard deviation for the 10-fold cross-validation, for each pair feature selection (in rows) and ML classifier method (in columns). The combinations with the highest AUC were Lasso + SVM, Lasso + RF, Lasso + KNN, and Lasso + NN with 0.93 ± 0.06, 0.92 ± 0.03, 0.92 ± 0.06, and 0.90 ± 0.05, respectively. On average, the SVM classifiers and Lasso feature selection methods had higher performance, 0.85 ± 0.05 and 0.86 ± 0.07, respectively. The ROC curves of the 10-fold cross-validation for Lasso + SVM and Lasso + NN are plotted in Figure 4.

Table 6 and Table 7 show, respectively, the AUC and ACC for the pair feature selection (in rows) and ML classifier method (in columns). The highest predictive performances, in terms of AUC, were obtained by the selection method Lasso + NN, MI + RF, Lasso + SVM, and WT + SVM with 0.90, 0.87, 0.86, and 84, respectively. On average, both SVM and Lasso methods had the mean higher performance in terms of AUC (0.82 ± 0.03 and 0.81 ± 0.06, respectively). In terms of ACC, MI + RF, and Lasso + SVM had the better performance, 0.85 and 0.76, respectively. Likewise, SVM and MI have on average the best performance with 0.72 ± 0.02 and 0.72 ± 0.07, respectively.

The ROC curve for the four models with higher AUC (i.e., Lasso + NN, MI + RF, Lasso + SVM, and WT + SVM) are shown in Figure 5. The result of the DeLong test showed that the AUC of these ROC curves was not significantly different (*p*-value > 0.5) as is shown in Figure 6. The AUC values, confident intervals of 95%, and the statistical significance p of each ROC curve were 0.87(0.74–0.95, *p* < 0.0001), 0.86(0.73–0.95, *p* < 0.0001), 0.90(0.77–0.97, *p* < 0.0001), and 0.84(0.70–0.93, *p* < 0.0001) for each model, respectively.

## 4. Discussion

Currently, several therapeutic alternatives are available to treat metastatic breast cancer [2]. However, the existence of multiple possibilities also requires a judicious assessment of the clinical response to the treatment administered to avoid unnecessary side effects, especially when it is not working adequately, allowing an early change to other potentially better therapeutic options. [^18^F]F-FDG PET/CT can evaluate globally the response to the treatment but can also identify individually treatment-resistant lesions, which harm the patient outcomes [32]. In this study, we showed that the prediction of the metabolic response to the therapy of individual metastatic breast cancer lesions is possible by using predictive models based on radiomic image features, clinical variables, and combinations of feature selection and machine-learning methods. We believe that such kinds of predictions could help us to develop new methodologies for assessing the treatment response in patients with metastatic cancer, where the post-therapeutic response assessment frequently shows heterogeneous or dissociated responses (i.e., responding and nonresponding metastatic lesions coexisting), which is associated with patient outcomes [8] both in this tumor and in other malignancies [33]. This could have several clinical applications, such as making it feasible to create quantitative indices of the expected dissociated response after the treatment, or for personalizing of the therapy; for instance, if prior to the treatment a metastatic lesion is suitably classified as treatment-resistant, it could additionally receive a local therapy such as radiotherapy. In this regard, a further objective of this study will be to use the predictions for individual metastatic breast cancer lesions to create predictive models of the global clinical response to the available therapies, which could allow choosing therapeutic regimens with the greatest probability of success and fewer adverse effects. However, for this purpose it is necessary to use only those predictive models, features, and clinical variables that achieve the highest performance, for which it is also necessary to know which combinations between the feature selection method and the machine-learning classifier are the msot suitable.

The emerging field of radiomics quantifies the phenotypic characteristics of tumor tissues on medical image features. Since [^18^F]F-FDG PET/CT is a valuable image method in oncology, commonly used to assess the tumor response to the treatment in breast cancer, we investigated the ability of radiomic features of [^18^F]F-FDG PET/CT along with which clinical information and ML algorithms to predict the metabolic response of metastatic breast lesions to the systemic treatment. To this aim, we developed and validated 49 predictive models, each one with different combinations of feature selection and ML methods. The most relevant set of features and/or clinical variables of each selection method, as well as the best hyperparameters of each ML classifier, were used. Finally, the model performances were assessed by 10-fold cross-validation and validation in the training and testing cohort, respectively. For this, the AUC and ACC were used as metrics. In short, we not only tried to predict the metabolic response but also to find an optimal configuration of feature selection and ML method for this specific clinical setting.

The combinations with the highest validation performance were the Lasso features selection method + NN and SVM classifiers, and MI selection method + RF classifier, each one with an AUC/ACC of 0.90/0.74, 0.86/0.76, and 0.85/0.87, respectively. Likewise, the mean AUC in the cross-validation was 0.90 ± 0.05, 0.93 ± 0.06, and 0.80 ± 0.08, respectively. It is important to mention that these methods have already shown their importance and robustness in other studies [25,28]. However, other combinations also showed comparably good performance so they should not be simply discarded, but the ML naive Bayes classifier performed poorly with all selection methods, achieving up to 30 percent less assertiveness than the best performance. Therefore, we would not recommend it for the prediction of the response to treatment in this class of patients.

In general, our results show that a radiomic approach, by using ML models, might be able to predict the tumor metabolic response to systemic treatments in patients with recurrent or metastatic breast cancer. However, from Table 4, we can appreciate that several clinical parameters as well as simple metabolic values, such as ΔER, PR, SUVmax, and SUVpeak which we selected for the predictions, were even well-ranked predictors., i.e., had a greater predictive significance than some image features. This shows that these parameters should always be taken into account for the construction of reliable predictive models, because of their well-known importance. Most of the PET/CT studies for the prediction of the treatment response in breast cancer do not include radiomic analysis, and they have been performed in a neoadjuvant context [34,35]; only a few studies consider the treatment response in patients with metastatic cancer [6,36]. On the other hand, concerning the existing PET/CT radiomic studies in breast cancer, they have only considered a neoadjuvant context [37,38,39,40]. To our best knowledge, no radiomic studies have been used along with ML to predict the treatment response in metastatic breast cancer, especially in a lesion-based approach. It is recognized that the performance of predictive models based on radiomics and ML is influenced by the feature selection method and the ML classifier chosen. Because different combinations have different performances, and it depends possibly on the tumor and clinical setting, some authors have recommended conducting performance comparisons of different combinations, for each tumor and/or clinical context [29]. The identification of an optimal ML method for radiomic applications is a crucial step towards stable and clinically relevant radiomic biomarker construction. We consider the importance of our study because of the lack of an ML-based radiomic approach to the assessment of metabolic response in patients with metastatic cancer.

In other pathologies and clinical contexts, several combinations of selection methods and ML classifiers are suitable for classification or prognostic purposes. For example, Du et al. [25] found the cross-combination Fisher score (FSCR) + KNN, SVM, or RF to be suitable for differentiation between recurrence and inflammation (AUC of 0.883, 0.867, and 0.892, respectively), by using [^18^F]F-FDG PET/CT images of patients treated for nasopharyngeal carcinoma. In addition to FSCR, they used other feature selection methods such as mutual information maximization, Relief-F, conditional mutual information maximization, minimum redundancy maximum relevance, and joint mutual information. Parmar et al. [28] investigated fourteen feature selection methods and twelve classifiers, in terms of their performance in predicting overall survival in patients with nonsmall-cell lung cancer (NSCLC). They used CT images and found that the Wilcoxon-test-based feature selection method and RF classification had the highest performance (AUC of 0.65 ± 0.02 and 0.66 ± 0.03, respectively). Yin et al. [24] aimed to identify optimal machine-learning methods for the preoperative differentiation of sacral chordoma and sacral giant cell tumors based on 3D nonenhanced computed tomography (CT) and CT-enhanced (CTE) features. They found that based on CTE features, the selection method Lasso plus the classifier, generalized linear models (GLM), had the highest AUC and ACC in the validating set, with 0.984 and 0.897, respectively. For CT features, RF + GLM had the highest AUC of 0.889. They demonstrated that CTE features performed better than CT features.

There are some limitations to this study. Firstly, it is a retrospective study with a relatively small cohort and heterogeneous group of patients; is regarding clinical, pathological features, and administered treatment; and pertains to only one institution. Thus, to improve the reproducibility and generalizability of this study, we used IBSI-based standardized radiomic features [21], which were normalized with the z-score method. However, a prospective multicenter study with a larger cohort of patients is necessary to confirm our results and improve the reliability and clinical applicability of this radiomic study. Equally, even when the tumor heterogeneity in breast cancer [41] makes it reasonable to consider each metastatic lesion as unique, it is not possible to be sure of the internal validity of our study, because we did not split the patients to have a training and test cohort at the patient’s level. It will be necessary to explore this aspect in future studies. On the other hand, we only compared seven commonly used feature selection methods and seven classification methods regarding their performance to predict metabolic response in patients with recurrent or metastatic breast cancer. There are many other methods, and therefore, possible combinations; we cannot state that we have found the most suitable combination. Hyperparameters cannot be learned by the algorithm directly during the training; rather, they must be set up before the training starts. In this study, four ML methods (LR, GNB, AdaBoost, and NN) were used with their default settings, whereas a hyperparameter tuning was performed for SVM, RF, and KNN, which might have resulted in enhanced performance of these last three methods. Likewise, since radiomic analysis uses a high amount of quantitative image features to create prediction models, the generalizability of these models is considered dependent on the robustness of selected features to construct them. Because of that, robustness studies investigating the stability of these features to the tumor delineation uncertainty, different image preprocessing methods, and reconstruction parameters have been performed [42,43,44]. One disadvantage of our study is that, due to its retrospective condition as well as image acquisition in a routinely clinical environment, we were unable to carry out such studies. However, more recently, Oliveira et al. [45] have shown that prediction models with robust feature preselection could be unsuccessful, indicating the need for a standardized imaging protocol acquisition and reconstruction.

In addition, patients included our study were in different lines of treatment, according to the evolutive stage of the disease, which harm the interpatient comparability of the metabolic response of the metastatic lesions. Despite all this, we have been able to predict the metabolic response to the treatment of metastatic breast cancer lesions with a significant AUC and ACC.

We believe that by recruiting a more homogeneous group of patients (i.e., comparable evolutive stages of the metasttic disease), segregating the patients into groups of comparable therapeutic regimes, the performance of our predictive models would be improved. Additionally, by doing a long-term follow-up of the patients, such models could be assessed regarding their ability to predict outcomes such as time free of disease and survival.

## 5. Conclusions

In conclusion, we constructed predictive models comprised of combinations between a feature selection method and an ML classifier, and feeding by radiomic [^18^F]F-FDG PET/CT features and clinical variables, to predict the metabolic response of individual metastatic breast cancer lesions in patients who underwent different therapies. In the validation cohort, the model Lasso + NN had the highest performance in terms of AUC, whereas MI + RF had it in terms of ACC. On average, Lasso and SVM methods had the best mean performance, although other combinations also showed high diagnostic performance. This comparative investigation may be an important reference for identifying effective combinations of a feature selection method and an ML classifier for reliable prognostication in this clinical setting.

## Figures and Tables

**Figure 1 cancers-14-02922-f001:**
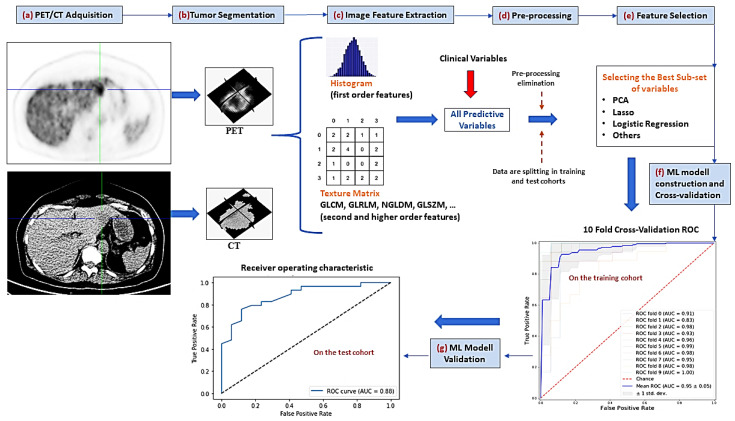
Radiomic pipeline process. (**a**) the first step consisted of PET/CT image acquisition (**b**) the tumor volume of interest is semi-automatically segmented (**c**) several first order and textural image features are extracted (**d**) a preprocessing step allowed to eliminate those highly correlated variables or with small variance (**e**) feature selection methods are applied to reduce the number of predictive variables (**f**) different machine-learning (ML) classifiers are used to construct the predictive models by using only the training cohort. Hyperparameter adjustment of these ML methods is performed by cross-validation (**g**) performance evaluation of predictive models by AUC-ROC analysis on the test cohort. PCA = principal component analysis, GLCM = gray-level co-occurrence matrix, GLRLM = gray-level run length matrix, NGLDM = neighboring gray-level dependence, GLSZM = gray-level size zone matrix, AUC = area under the curve, ROC = receiver-operating characteristic.

**Figure 2 cancers-14-02922-f002:**
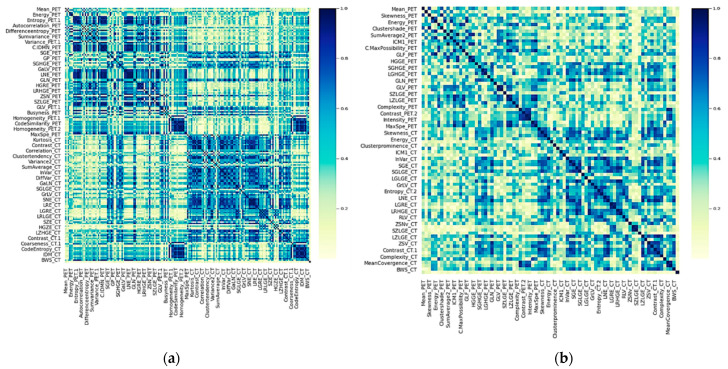
Heatmap of the absolute coefficient of Spearman’s correlations between clinical and imaging features. Darker blue colors indicate a higher correlation. (**a**) Before preprocessing and (**b**) After preprocessing, which consisted of elimination of one of two highly correlated variables.

**Figure 3 cancers-14-02922-f003:**
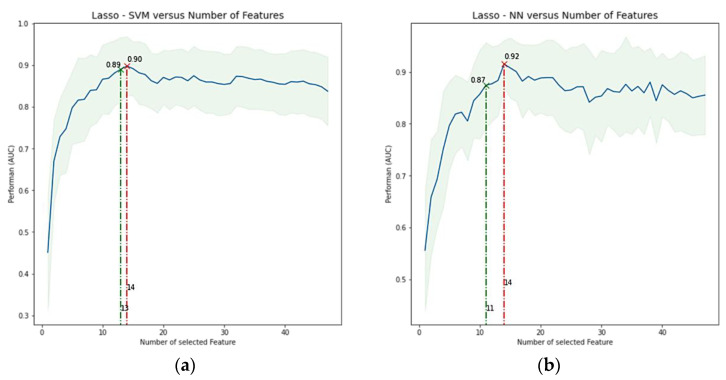
Average performance in terms of AUC ± SD versus number of features included. (**a**) Lasso + SVM model, (**b**) Lasso + NN model. The highest AUC for each model was found with 14 features. However, a more optimized feature number, i.e., a smaller number of features but with only a small change concerning the maximal value, was found with 13 features in a, but not in b (changes < 0.05 were allowed if there were a significant reduction in the number of features). Lasso = least absolute shrinkage and selection operator, SVM = support vector machine, RF = random forest, AUC = area under the receiver operator characteristic curve, and SD = standard deviation.

**Figure 4 cancers-14-02922-f004:**
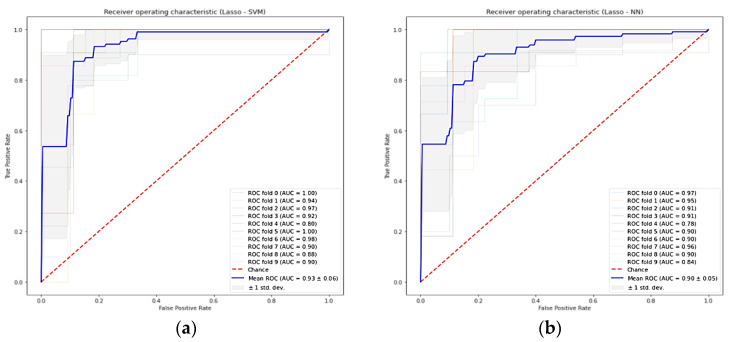
Receiver-operating characteristic curve for the ten-fold cross-validation of predictive models in the training cohort. (**a**) Lasso + SVM model, (**b**) Lasso + NN model. Lasso = least absolute shrinkage and selection operator, SVM = support vector machine, RF = random forest, AUC = area under the receiver operator characteristic curve, and standard deviation (std dev).

**Figure 5 cancers-14-02922-f005:**
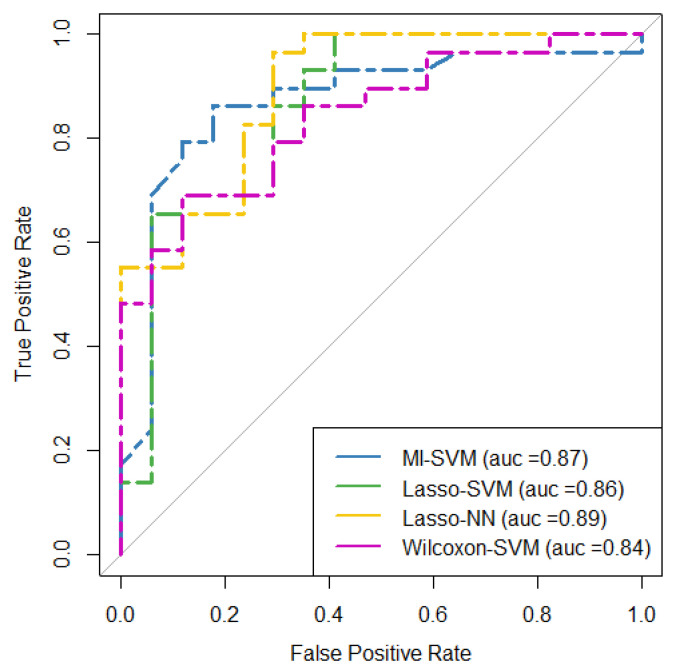
ROC curves for the four predictive models with the highest performance in terms of AUC in the testing cohort. MI = mutual information, SVM = support vector machine, Lasso = absolute shrinkage and selection operator, NN = neural network, AUC = area under the receiver operator characteristic curve, ROC = receiver-operating characteristic.

**Figure 6 cancers-14-02922-f006:**
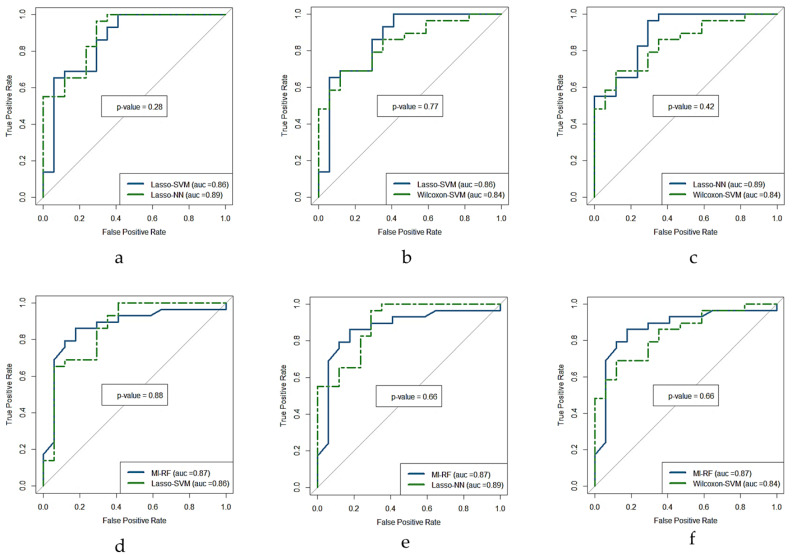
DeLong test ROC curves of the four predictive models with highest performance in terms of AUC in the test cohort. (**a**) Lasso + SVM versus Lasso + NN, (**b**) Lasso + SVM versus Wilcoxon + SVM, (**c**) Lasso + NN versus Wilcoxon + SVM, (**d**) MI + RF versus Lasso + SVM, (**e**) MI + RF versus Lasso + NN, and (**f**) MI + RF versus Wilcoxon + SVM, where *p* is the statistical significance (if <0.5 is significant). MI = mutual information, Lasso = absolute shrinkage and selection operator, SVM = support vector machine, RF = random forest, NN = neural network, AUC = area under the receiver operator characteristic curve, ROC = receiver-operating characteristic.

**Table 1 cancers-14-02922-t001:** Feature selection and classification methods that were used to obtain the predictive models.

Feature Selection Method	ML Classifier
AFT (ANOVA-F-test)	SVM (support vector machine)
MI (mutual information)	GNB (Gaussian naive Bayes)
PCA (principal component analysis)	RF (random forest)
ICA (independent component analysis)	LR (logistic regression)
Lasso (least absolute shrinkage and selection operator)	KNN (k-nearest neighborhood)
CL (clustering)	AdaBoost (adaptive boosting)
WT (Wilcoxon test)	NN (neural network)

**Table 2 cancers-14-02922-t002:** Patient demographic information and clinical characteristics at initial diagnosis.

Characteristic	Number	Percentage
Total patients	48 (mean age 48.1 years)	100
Affected side		
right	26	54.2
left	22	45.8
Histologic type		
ductal	42	87.5
lobular	5	10.4
other	1	2.1
Tumor size ^a^ (pT)		
T1a-b	12	25
T1c	15	31.3
T2	11	22.9
T3	5	10.4
Nodal affectation ^a^ (pN)		
N0	14	29.2
N1	22	45.8
N2a-b	4	8.3
N3a	2	4.2
N3b	1	2.1
Mestatase ^a,b^ (M)		
M0	20	39.6
M1	1	2.1
Mx	22	43.8
TNM clinical stage ^a^		
IA	13	27.1
IB	0	0
IIA	16	33.3
IIB	4	8.3
IIIA	6	12.5
IIIB	0	0
IIIC	3	6.3
IV	1	2.1
Estrogen receptor positivity		
negative	17	54.2
low	4	54.2
moderate	11	54.2
strong	16	54.2
Progesterone receptor positivity		
negative	24	50
low	8	16.7
moderate	7	14.62
strong	9	18.8
Her2 positivity		
0	33	68.8
1	15	31.3
Histologic grade ^c^		
well	1	2.1
moderate	20	41.7
poor	26	54.2

^a^ Five patients had no available information; ^b^ Twenty-two patients had no available or unknown M stage, and one had a de novo metastatic breast cancer. ^c^ One patient had no available information. pT and pN are the pathological T and N stage, respectively. Her2 = human epidermal growth factor receptor 2. Stage classification was according to the seventh edition of the AJCC Cancer Staging Manual [17].

**Table 3 cancers-14-02922-t003:** Patients’ treatment and affectations’ locations.

Patient	Treatment	Metastatic Lesions
1	ChT	Liver (1)
2	ChT, Xgeva, Zoladex, and RT	Bone (1)
3	ChT and RT	Liver (1), Lung (3), LN (3)
4	ChT	Bone (1), LN (3)
5	Taxotere and Parjeta	Liver (1), LN (3)
6	Taxol and Herceptin	Breast (1), LN (2)
7	Taxotere, Herceptin, Perjeta, and Xgeva	Breast (1), Bone (3), Liver (3). LN (3)
8	Navelbine	Bone (1), LN (6)
9	Taxol	LN (4)
10	ChT	Bone (1), Liver (7)
11	Taxotere, Herceptin, and Perjeta	Liver (3), LN (5)
12	Paclitaxel and Bevacizumab	LN (7)
13	ChT	Bone (8), LN (4)
14	Navelbine	LN (9)
15	ChT	Bone (1), Liver (2). Pleura (8)
16	Aromasin, Afinitor, Xgeva, and RT	Bone (6), LN (3)
17	Xeloda, Avastin, Bortezomib, and RT	Bone (4), LN (4)
18	ChT and RT	LN (2)
19	Liver Meta Excision, Xgeva, and Zometa	Bone (7), Liver (3)
20	Taxotere, Herceptin, and Perjeta	Bone (14), Liver (2), LN (1)
21	Letrozol, changed to Fulvestrant	Bone (3), LN (2)
22	Paclitaxel	Bone (3), Liver (2)
23	Arimidex and Herceptin	LN (1)
24	Lipidox, lung meta excision	LN (1)
25	Vinorelbine and Trastuzumab	Liver (1)
26	Arimidex, lung Meta excision	Bone (1), LN (2)
27	ChT and Trastuzumab	LN (2)
28	ChT and RT	Bone (1), LN (2)
29	Avastin and Abraxane	Bone (1), Suprarenal (1)
30	ChT	Bone (3)
31	ChT	LN (1)
32	ChT and liver metastase excision	Liver (3)
33	Epirubicin und Docetaxel	Liver (1)
34	Xgeva and RT	Bone (2)
35	Xgeva and RT	Bone (1)
36	Xvega	Bone (6)
37	Zometa	Bone (4)
38	Zometa and RT	LN (1)
39	ChT	Bone (2)
40	ChT and RT	Bone (1), Liver (1). Lung (1). LN (1)
41	Radioembolization	Liver (1), Spleen (1)
42	Taxotere and Avastin	LN (2)
43	Taxotere and Avastin	Bone (6), LN (2)
44	Gemzar, Cisplatin, and Avastin	LN (3)
45	Taxol and Xgeva	Breast (1), Bone (6)
46	Trastuzumab and Xgeva	Bone (3), Spleen (4)
47	Xeloda and RT	Liver (2)
48	Methotrexate and Xgeva	Bone (1), LN (4)

ChT = Chemotherapy (exact regimen not available); RT: radiation therapy; LN = lymph node. The values in parentheses indicate the number of lesions that were segmented for that patient.

**Table 4 cancers-14-02922-t004:** Ranked predictors of the four models with higher AUC in the validation.

Model	Number of Predictors	Ranked Predictors (Predictors on the Left Are of Greater Predictive Significance)
Lasso + SVM and Lasso + NN	14	BWS-PET, SUVmax, Skewness-CT, Kurtosis-PET, ΔER, PR, T, Her2neu-Metastasis, PR-Metastasis, Affectation-Side, ΔGrading, Her2neu-Primary, P53
MI + RF	13	LGHGE-CT, PR, Energy-ET, Correlation-PET, Max-PET, ER, SUVpeak, ZSNv-CT, SRLGE-CT, Age at Diagnosis, BWS-PET, PR-Metastasis, P53
Wilcoxon + SVM	59	PR, LGHGE-CT, Variance-T, GLN-CT, Correlation-PET, GLV-CT, GLV-PET, ER-Mestastasis, ZSNv-CT, BWS-PET …

PR = progesterone receptor status, ER = estrogen receptor status, Her2 = human epidermal growth factor receptor 2, grading = histologic grade, P53 = p53 expression intensity, T = tumor size (pathological T), SUVmax and SUVpeak = maximum and peak standardized uptake values, respectively, BWS = black white symmetry, LGHGE = long-gap high-gray-level emphasis, Max = maximum intensity, ZSNv = zone-size nonuniformity, SRLGE = short-run low-gray-level emphasis, GLN = gray-level nonuniformity, GLV = gray-level variance. Δ means change in a pathologic marker between primary and the metastatic lesions. MI = mutual information, SVM = support vector machine, Lasso = absolute shrinkage and selection operator, NN = neural network, and RF = random forest. For Wilcoxon + SVM, only the ten most significant predictors are listed.

**Table 5 cancers-14-02922-t005:** Cross-validation performance of all predictive models (FS + Classifier) in terms of the AUC. Values indicate the AUC ± SD of each model or mean AUC ± one SD for a determined feature selection method or classifier. All models were evaluated on the validation cohort.

		Classifier							
	Model	SVM	GNB	RF	LR	KNN	Ada Boost	NN	Mean FS
Feature selection (FS)	AFT	0.83 ± 0.06	0.78 ± 0.08	0.76 ± 0.08	0.74 ± 0.07	0.78 ± 0.12	0.80 ± 0.08	0.78 ± 0.08	0.78 ± 0.03
	MI	0.80 ± 0.10	0.78 ± 0.10	0.80 ± 0.08	0.76 ± 0.08	0.86 ± 0.08	0.75 ± 0.06	0.78 ± 0.06	0.79 ± 0.04
	PCA	0.84 ± 0.08	0.79 ± 0.07	0.81 ± 0.07	0.71 ± 0.08	0.75 ± 0.11	0.68 ± 0.13	0.79 ± 0.07	0.77 ± 0.06
	ICA	0.88 ± 0.08	0.75 ± 0.05	0.75 ± 0.09	0.73 ± 0.04	0.73 ± 0.12	0.64 ± 0.09	0.74 ± 0.08	0.75 ± 0.07
	Lasso	0.93 ± 0.06	0.80 ± 0.10	0.92 ± 0.03	0.77 ± 0.08	0.92 ± 0.06	0.79 ± 0.13	0.90 ± 0.05	0.86 ± 0.07
	CL	0.80 ± 0.15	0.71 ± 0.08	0.86 ± 0.08	0.73 ± 0.10	0.77 ± 0.10	0.78 ± 0.10	0.75 ± 0.09	0.77 ± 0.05
	WT	0.84 ± 0.06	0.75 ± 0.08	0.76 ± 0.09	0.75 ± 0.09	0.82 ± 0.09	0.80 ± 0.09	0.79 ± 0.06	0.79 ± 0.04
Mean Classifier		0.85 ± 0.05	0.77 ± 0.03	0.81 ± 0.06	0.74 ± 0.02	0.80 ± 0.07	0.75 ± 0.06	0.79 ± 0.05	

AUC = area under the receiver operator characteristic curve, SD = standard deviation, AFT = ANOVA F-test, MI = mutual information, PCA = principal component analysis, ICA = independent component analysis, Lasso = least absolute shrinkage and selection operator, SVM = support vector machine, GNB = Gaussian naive Bayes, RF = random forest, LR = logistic regression, WT = Wilcoxon test, NN = neural network, KNN = k-nearest neighborhood, CL = clustering, AdaBoost = adaptive boosting.

**Table 6 cancers-14-02922-t006:** Validation performance of all predictive models (FS + Classifier) in terms of the AUC. Values indicate the AUC of each model or mean AUC ± one SD for a determined feature selection method or classifier. All models were evaluated on the validation cohort.

		Classifier							
	Model	SVM	GNB	RF	LR	KNN	Ada Boost	NN	Mean FS
Feature selection (FS)	AFT	0.78	0.70	0.77	0.76	0.80	0.72	0.82	0.76 ± 0.04
	MI	0.79	0.79	0.87	0.69	0.78	0.74	0.81	0.78 ± 0.06
	PCA	0.80	0.69	0.81	0.68	0.65	0.71	0.7	0.72 ± 0.06
	ICA	0.83	0.66	0.74	0.63	0.72	0.71	0.76	0.72 ± 0.07
	Lasso	0.86	0.70	0.83	0.78	0.83	0.77	0.90	0.81 ± 0.07
	CL	0.81	0.78	0.77	0.79	0.81	0.65	0.80	0.77 ± 0.06
	WT	0.84	0.75	0.74	0.73	0.71	0.74	0.79	0.76 ± 0.04
Mean Classifier		0.82 ± 0.03	0.72 ± 0.05	0.79 ± 0.05	0.72 ± 0.06	0.76 ± 0.07	0.72 ± 0.04	0.80 ± 0.06	

AUC = area under the receiver operator characteristic curve, SD = standard deviation, AFT = ANOVA F-test, MI = mutual information, PCA = principal component analysis, ICA = independent component analysis, Lasso = least absolute shrinkage and selection operator, SVM = support vector machine, GNB = Gaussian naive Bayes, RF = random forest, LR = logistic regression, WT = Wilcoxon test, NN = neural network, KNN = k-nearest neighborhood, CL = clustering, AdaBoost = adaptive boosting.

**Table 7 cancers-14-02922-t007:** Validation performance of all predictive models (FS + Classifier) in terms of the ACC. Values indicate the ACC of each model, or mean ACC ± one SD for a determined feature selection method or classifier. All models were evaluated on the validation cohort.

		Classifier							
	Model	SVM	GNB	RF	LR	KNN	Ada Boost	NN	Mean FS
Feature selection (FS)	AFT	0.72	0.72	0.67	0.70	0.70	0.70	0.70	0.70 ± 0.02
	MI	0.70	0.63	0.85	0.67	0.72	0.74	0.72	0.72 ± 0.07
	PCA	0.74	0.63	0.74	0.70	0.63	0.70	0.63	0.68 ± 0.05
	ICA	0.72	0.57	0.61	0.61	0.72	0.72	0.74	0.67 ± 0.07
	Lasso	0.76	0.54	0.74	0.72	0.74	0.65	0.72	0.70 ± 0.08
	CL	0.72	0.74	0.67	0.72	0.72	0.65	0.74	0.71 ± 0.03
	WT	0.70	0.63	0.67	0.72	0.59	0.50	0.70	0.64 ± 0.08
Mean Classifier		0.72 ± 0.02	0.64 ± 0.07	0.71 ± 0.08	0.69 ± 0.04	0.69 ± 0.06	0.67 ± 0.08	0.71 ± 0.04	

AUC = area under the receiver operator characteristic curve, SD = standard deviation, AFT = ANOVA F-test, MI = mutual information, PCA = principal component analysis, ICA = independent component analysis, Lasso = least absolute shrinkage and selection operator, SVM = support vector machine, GNB = Gaussian naive Bayes, RF = random forest, LR = logistic regression, WT = Wilcoxon test, NN = neural network, KNN = k-nearest neighborhood, CL = clustering, AdaBoost = adaptive boosting.

## Data Availability

Data are available on: https://dataverse.harvard.edu/dataset.xhtml?persistentId=doi:10.7910/DVN/LA8PGN (accessed on 12 January 2022).

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
