# Peer review of "Analysis of Cross-Combinations of Feature Selection and Machine-Learning Classification Methods Based on [18F]F-FDG PET/CT Radiomic Features for Metabolic Response Prediction of Metastatic Breast Cancer Lesions"

_cancers, 2022, doi:10.3390/cancers14122922_

Round 1
Reviewer 1 Report
This study dealt with the machine learning models of FDG PET/CT for predicting metabolic response in metastatic breast cancer. Although this is a novel and interesting topic, there are many critical issues in the Introduction, Methods and Results.
Introduction
1) It is difficult understand the clinical need for predicting metabolic response by using pretherapeutic FDG PET/CT and machine learning in metastatic breast cancer. Still, the RECIST is standard method for therapy response in metastatic breast cancer. Predicting RECIST or prognosis is more appropriate outcome in these patients.
Materials and Methods
2.1. Patient Cohort
2) Medical 120 records, as well as pathologic and radiologic reports, were reviewed to identify a set of 121 patients with metastasis, who received chemotherapy or target therapy and hat basal and 122 follow-up [18F]F-FDG) PET/CT.
; It is difficult to understand clinical setting of the finally included patients. In other words, patients with initially diagnosed breast cancer without previous therapy or patients with recurrent or persistent breast cancer with a history of previous therapy.
2.2. PET/CT image acquisition
3) It is necessary to describe the FDG uptake time and PET reconstruction method.
2.3. ROI Delineation
4) It is necessary to describe the tumor segmentation method.
5) It is necessary to describe the VOI targets for the therapy response clearly: primary tumor, regional lymph node or distant metastatic sites.
6) It is necessary to clarity whether pretherapy PET/CT only used for the machine learning.
2.4. Image Preprocessing
7) PET image intensity was converted to SUVs. To remove individual acquisition differences, and in-line with other radiomic studies [16], the images were normalized and resampled into a voxel size of 1 × 1 × 1 mm3. Further image preprocessing was not performed.
; It is difficult to understand this normalization and resampling process, because it seems that the same kind of PET/CT scan and image acquisition protocol. Especially, this preprocessing cannot be applied for the PERCIST.
2.6. PET/CT response assessment
8) It is difficult to understand the use of PERCIST as primary outcome. Because still RECIST is standard method for the response evaluation in metastatic breast cancer, just predicting metabolic response in these patients had low clinical value.
Results
9) In the Table 2, initial overall stage should be included.
10) In the Table 2, it is necessary to clarify whether the TNM stages were clinical or pathological.
11) In the Table 2, it is difficult to understand why there were 44% of patients with MX. In this study, all subjects underwent PET/CT before therapy. In this setting, clinical M stage could be determined.
12) It is necessary to describe the version of the TNM staging system.
13) In the Table 3 and 4, the statistical comparisons between the AUCs should be performed.
14) It is necessary to compare conventional PET parameters such as SUV, MTV and TLG with the machine learning models, because the conventional PET parameters are more simple and clinically applicable.
Discussion
15) Lack of true internal validation cohort is another main limitation of this study.
Author Response
Thank you very much for your comments.
Please see attached file

Reviewer 2 Report
The authors write a fascinating and didactic work trying to identify optimal ML methods-models for clinically relevant radiomic 18F-FDG PET-CT biomarkers construction in metastatic breast cancer patients.
Please modify Line 122: had or have instead of hat; Line 179: 101 instead of 110.
It could be better to include Table S3. Patient´s treatment and affectation’s places, in the paper.
In the conclusions paragraph, it could be better to modify "systemic therapy" in "patients undergoing different therapies" (different chemotherapy regimens and/or radiotherapy -see patients 2;3;16;17;18;34;35 and 40- ).
Round 2
Reviewer 1 Report
Although methodological issues were resolved well, there is a still fundamental issue regarding clinical relevance of the study design and results. Although there were several previous studies regarding the advantages of PERCIST over RECIST in metastatic breast cancer, still RECIST is the standard method for response evaluation. Based on the presented results, still it is difficult to omit follow-up PET after therapy. For example, new lesion detection. Also, it difficult to determine the therapeutic plan based on the presented results. Therefore, still clinical value of this study is low.